# Boldo Restores Vascularization and Reduces Skeletal Muscle Inflammation in Symptomatic Mice with Dysferlinopathy

**DOI:** 10.3390/ijms26209945

**Published:** 2025-10-13

**Authors:** Walter Vásquez, Felipe Troncoso, Andrea Lira, Carlos Escudero, Juan C. Sáez

**Affiliations:** 1Departamento de Fisiología, Pontificia Universidad Católica de Chile, Santiago 8331150, Chile; wvasqueza@uc.cl; 2Laboratorio de Fisiología Vascular, Departamento de Ciencias, Universidad del Bio-Bio, Chillán 3800708, Chile; 3Departamento de Morfología, Facultad de Medicina, Universidad Andres Bello, Santiago 8370211, Chile; 4Group of Research and Innovation in Vascular Health, GRIVAS Health, Chillán 3800708, Chile; 5Instituto de Neurociencias, Centro Interdisciplinario de Neurociencias de Valparaíso, Universidad de Valparaíso, Valparaíso 2360102, Chile

**Keywords:** dysferlinopathy, peumus boldus, skeletal muscle

## Abstract

Dysferlinopathies are progressive muscular dystrophies caused by DYSF mutations, leading to impaired membrane repair, chronic inflammation, lipid accumulation, and muscle degeneration. No approved therapies currently halt the progression of this disease. Here, we evaluated the effects of daily oral administration of pulverized Boldo (*Peumus boldus*) leaves, commonly used as a nutraceutical, to blAJ mice, a model of dysferlinopathy. Symptomatic bIAJ mice were treated for four weeks with Boldo and presented significantly improved grip strength and restored endothelial-dependent vasodilation. Muscle perfusion and capillary density in the gastrocnemius were both enhanced by treatment. Histological analyses revealed that Boldo prevented myofiber atrophy, reduced centrally nucleated fibers, and improved muscle tissue architecture. Lipid accumulation observed in blAJ muscles was absent in Boldo-treated mice. At the cellular level, Boldo normalized sarcolemma membrane permeability (dye uptake) and reduced mRNA levels of inflammasome components (NLRP3, ASC, and IL-1β), suggesting anti-inflammatory activity. These findings indicate that Boldo improves vascular and muscle integrity, supporting its potential as a complementary therapeutic strategy for dysferlinopathy.

## 1. Introduction

Dysferlin is a 237 kDa membrane protein belonging to the ferlin family and is composed of seven C2 domains, a DysF domain, and two Fer domains [1,2]. It is predominantly expressed in the sarcolemma and T-tubules of striated muscles [3,4]. It plays key roles in cell membrane repair [1] and membrane protein trafficking [5], affecting vesicle fusion [6], biogenesis of T-tubules, Ca^2+^ signaling [7], phagocytic processes [8], endothelial cell adhesion, repair, and angiogenesis [9,10]. Indeed, dysferlin plays a critical role in repairing the endothelial cell membrane by facilitating its fusion with lysosomes, suggesting its active involvement in recovery processes and maintenance of cellular integrity in the endothelium [11]. Specifically, dysferlin regulates endothelial cell adhesion by preventing proteasomal degradation of PECAM-1/CD31, a key adhesion protein for angiogenesis and vascular integrity. Loss of dysferlin impairs endothelial adhesion and angiogenic responses in vivo, suggesting a role in maintaining capillary networks and vascular function [9].

Mutations in the *DYSF* gene cause a set of muscular dystrophies collectively known as dysferlinopathies, including limb–girdle muscular dystrophy type 2B (LGMDR2) and Miyoshi myopathy (MM) [10,12,13]. Alterations in the *DYSF* gene are related to a reduction in dysferlin expression and seem to affect muscle fibers in the ability to repair themselves after sarcolemma damage [14]. This process leads to persistent inflammation, muscle degeneration, and the progressive replacement of muscle tissue by fat [15].

Clinically, dysferlinopathies are characterized by a progressive, generally slow decline of the appendicular system’s motor function, which usually manifests during adolescence [10,16]. Symptoms are highly variable among individuals, although they tend to develop certain common patterns. Regardless of the appearance of signs of muscle weakness, increases in serum creatine kinase (CK) levels are typical and indicative of muscle damage. This has been used for years as a diagnostic criterion in the presence of motor impairment [17].

The presence of distinctive infiltrates of inflammatory cells in skeletal muscle biopsies from patients with dysferlinopathy has been described in the literature, characterized by the presence of macrophages, along with minimal expressions of MHC class I molecules (major histocompatibility complex class I) and significant deposition of the complement complex C5b-9, which distinguishes this profile from other inflammatory pathologies [18]. Other studies reported variability in muscle fiber size, confirming the presence of necrotic fibers, regenerative fibers [19,20], as well as increases in endomysia and perimysium connective tissue and fat accumulation [21].

The leaves of the Chilean edemic tree *Peumus boldus*, known among locals as ¨Boldo¨, contain alkaloids, flavonoids, phenolic acids, and essential oils conferring antioxidant and anti-inflammatory properties [22,23,24]. Boldine, an active alkaloid present in Boldo leaves, has shown protective effects in animal inflammatory models [25,26,27], potentially through the blockade of Cx and pannexin-1 hemichannels (Cx HCs and Panx1 HCs, respectively) [28,29]. Boldo is used as a nutraceutical in several countries, and its beneficial effects are well known but require scientific support.

Additionally, the expression of hemichannels comprised of connexins has been observed in skeletal muscle fibers of blAJ mice (dysferlinopathy model), and has been associated with increased sarcolemma permeability and elevated basal intracellular Ca^2+^ signaling [30]. In human skeletal muscles affected by dysferlinopathy, increases in the expression of hemichannels composed of Cx40.1, Cx43, and Cx45 isoforms located in the sarcolemma have been described [31]. Treatment with boldine normalizes the differentiation fate of myoblasts as well as the altered features of skeletal muscles from bIAJ mice [30]. De novo Cx HC expression contributes to the factors causing elevated basal intracellular Ca^2+^ levels observed in dysferlin-deficient myocytes, and is sufficient to disrupt their cellular homeostasis. Therefore, a direct relationship between the absence of dysferlin and the induction of connexin expression has been suggested [31].

Currently, there are no approved therapies that can halt or reverse the progression of dysferlinopathy. Given the multifaceted roles of dysferlin in muscle membrane repair, endothelial adhesion, and angiogenesis, our study aimed to determine whether treatment with Boldo, commonly used as a nutraceutical, improves muscle function and vascular perfusion, ultimately reducing inflammation and lipid infiltration in blAJ mice.

## 2. Results

### 2.1. Boldo Improves Grip Strength and Blood Perfusion in Dysferlinopathy Mice

Dysferlinopathy leads to a gradual decline in muscle strength [32]. We analyzed the effects of daily oral treatment with Boldo for one month and evaluated physical performance using the four-limb suspension test in blA/J mice. Before the intervention, blAJ mice already showed reduced strength compared to WT controls (mean pre-test time: 126.5 ± 17.3 s vs. 195.8 ± 17.1 s, respectively, *n* = 4; *p* < 0.01). WT mice maintained stable performance after one month (post: 223.0 ± 9.7 s), while untreated blAJ mice exhibited a further decline (post: 116.5 ± 17.2 s). In contrast, Boldo-treated blAJ mice showed improved performance (post: 188.0 ± 17.1 s), surpassing their pre-treatment values (139.3 ± 18.3 s). At the end of the treatment period, WT mice performed significantly better than blAJ mice (*p* < 0.01), and Boldo-treated blAJ mice showed significant improvement compared to untreated blAJ mice (*p* < 0.05). These results confirm the progressive functional decline in dysferlin-deficient mice and demonstrate that Boldo treatment provides a partial, but significant, recovery of muscle strength (Figure 1).

To determine whether the decline in muscle strength observed in blAJ mice is associated with impaired muscle perfusion, we assessed microvascular blood flow in the gastrocnemius muscle with laser Doppler imaging (Figure 2A,B). The blAJ mice exhibited significantly reduced basal perfusion compared to wild-type controls, an effect that was reversed upon treatment with Boldo, as shown in the time-course analysis of perfusion traces (Figure 2B) and in the quantification of average perfusion units (Figure 2C).

To explore whether endothelial dysfunction underlies the perfusion deficit, we evaluated vasodilatory responses to acetylcholine, a classic endothelium-dependent vasodilator. As expected, acetylcholine increased perfusion by approximately 80% over the baseline in wild-type mice (Figure 2C). In contrast, this response was markedly blunted in blAJ mice, suggesting impaired endothelial function. Notably, Boldo treatment restored acetylcholine-induced vasodilation in blAJ mice, reaching values similar to the WT mice, supporting a functional recovery of endothelial responsiveness (Figure 2C).

### 2.2. Boldo Increases Capillary Density in Skeletal Muscles from blAJ Mice

To determine whether the improved blood perfusion observed after Boldo treatment was associated with structural changes in the vasculature, we evaluated the capillary density in gastrocnemius muscle sections using immunofluorescence for CD31^+^, an endothelial cell marker. Compatible with reduced microvascular perfusion, blAJ mice exhibited a reduced number of CD31^+^ capillaries per muscle fiber compared to WT mice. Muscles from Boldo-treated blAJ mice showed a partial recovery in capillary density, as evidenced by a higher number of CD31^+^ structures per fiber than in untreated blAJ mice (Figure 3B). These results suggest that Boldo may promote vascular remodeling in dysferlin-deficient muscle, potentially contributing to the observed improvements in perfusion and muscle function.

Given that loss of strength in blAJ mice is accompanied by muscle atrophy [33], we evaluated the cross-sectional area of myofibers in *tibialis anterior* muscles, and found that muscles from blAJ mice presented a clear reduction in CSA of myofibers (~40%) compared to control muscles (Figure 4B). This reduction was prevented by Boldo treatment as indicated by an increase (~34%) in myofiber cross-sectional area compared to the untreated blAJ mice (Figure 3B). Additionally, blAJ mice muscles presented a significantly higher number of muscle fibers with internal nuclei (~24%), which was less pronounced in the Boldo-treated group that had values close to those found in WT mice muscles (Figure 4C).

Considering that the accumulation of lipids within muscles is widely recognized as a hallmark of dysferlinopathy [34], we decided to evaluate whether Boldo can reverse this process. To this end, we assessed lipid accumulation in gastrocnemius muscles using oil red O staining. While positive staining was observed in muscle sections of blAJ mice, no lipid accumulation was detected in muscles from WT mice or blAJ mice treated with Boldo (Figure 5).

### 2.3. Boldo Improves the Muscular Architecture of blAJ Mice

Histological examination of tibialis anterior muscle sections stained with hematoxylin, eosin, and Masson’s trichrome revealed marked differences between groups (Figure 6A). WT mice exhibited uniform polygonal muscle fibers, peripheral nuclei, and intact sarcolemma borders with normal connective tissue distribution. In contrast, blAJ muscles showed classic dystrophic features, including pronounced fiber size variability, centrally located nuclei, vacuolization, and the presence of irregular sarcolemma projections resembling microvilli. These alterations were accompanied by increased cellularity, anisokaryosis, disorganized interstitial architecture, and accumulation of collagen, consistent with chronic degeneration and failed regeneration processes.

Muscles from Boldo-treated blAJ mice displayed partial restoration of tissue architecture. Fiber morphology was more homogeneous, with fewer vacuoles and reduced sarcolemma projections. The number of centrally nucleated fibers was lower, and the extracellular matrix appeared better organized with more regular collagen deposition. These changes suggest that Boldo treatment mitigates structural deterioration associated with dysferlin deficiency and supports muscle remodeling.

### 2.4. Boldo Reduces the Sarcolemma Permeability of Myofibers from blAJ Mice

Previous studies have demonstrated that myofibers from blAJ mice express large-pore channels permeable to ions and small molecules [30,33]. We decided to evaluate sarcolemma permeability in freshly isolated myofibers from FDB muscles. This was carried out by using the ethidium (Etd^+^) uptake assay leakage methods, which have been widely used to assess the functional state of undocked hemichannels in diverse cellular systems [35]. 

Fluorescence intensity of myofibers from blAJ J mice increased over time to a greater extent than that of myofibers from blAJ mice treated with Boldo, which overlapped with the intensity of myofibers from WT mice (Figure 7A). After five minutes of recording, the application of La^3+^ reduced the progressive increase in fluorescence intensity in all myofibers studied (Figure 7A). From these experiments, we found that the Etd^+^ rate of myofibers from blAJ mice (0.58 ± 0.08 AU/min) was significantly higher than that of the WT group (0.24 ± 0.02 AU/min) (Figure 7B). Importantly, myofibers from Boldo-treated blAJ animals presented dye uptake rate values similar to myofibers from WT mice (0.31 ± 0.04 AU/min) (Figure 7B).

### 2.5. Boldo Reduces Levels of Inflammatory Markers in Skeletal Muscles from blAJ Mice

Prior research examined muscle biopsies from control and dysferlin-deficient patients, demonstrating that inflammasome components are present and upregulated in muscles from dysferlin-deficient patients [36]. We evaluated some components of the inflammasome using qPCR and found that blAJ mice exhibited greater NLRP3 mRNA levels (2.65 ± 0.98) compared to wild-type mice (0.49 ± 0.37), a phenomenon that was prevented with Boldo treatment (0.99 ± 0.32) (Figure 8A). This same behavior was observed in relation to ASC mRNA levels (Figure 8B) (blAJ mice: 3.23 ± 1.20 and WT: 0.76 ± 0.41, Boldo-treated mice: 1.36 ± 0.79, respectively) and IL-1β mRNA levels (Figure 8C), which were significantly elevated in blAJ mice (6.64 ± 1.63) compared to WT (1.07 ± 0.61), and reduced in Boldo-treated blAJ mice (2.23 ± 0.99). Treatment with Boldo significantly diminished the expression of these pro-inflammatory markers, indicating its potential anti-inflammatory effects in dysferlinopathy.

## 3. Discussion

This study provides evidence supporting the use of pulverized Boldo leaves to improve physical performance (hanging test), blood perfusion, and vascularization, as well as reduce sarcolemma permeabilization, lipid accumulation, and inflammation in skeletal muscles from a murine dysferlinopathy model. Currently, this disease has no cure. Therefore, the treatment proposed here, which consists of ingesting leaves from a native plant that has no reported side effects and is accepted as a medicinal plant [37,38], could be a valuable contribution to existing recommended treatments.

Boldo is recognized as an herbal remedy in several pharmacopeias. Its main alkaloidal constituent, boldine [38], accounts for approximately 0.1% of the total alkaloid content and is considered the primary bioactive compound [39]. This alkaloid blocks large-pore channels, such as Cx43, Cx45, and Panx1 hemichannels, as well as the P2X7 receptor [29,40,41]. Recently, symptomatic blAJ mice treated with boldine were shown to present normal muscle performance [30]. Our study reports similar effects of Boldo-treatment, which improved physical performance in the hanging test. These findings raise the possibility that Boldo’s effects may be mediated by hemichannel blockade, given that dysferlinopathic mice deficient in Cx43 and Cx45 in skeletal muscles retain normal physical performance [33].

In addition to enhancing blood perfusion, we found that Boldo treatment partially restores capillary density in skeletal muscle of blAJ mice. Dysferlin deficiency has previously been associated with reduced capillary density and impaired angiogenic responses due to endothelial dysfunction and defective PECAM-1 stabilization [9]. We consistently observed a reduction in the number of CD31^+^ capillaries per fiber in blAJ mice, which was partially reversed by Boldo. This increase in capillary density suggests that Boldo may preserve or promote vascular integrity in dystrophic muscles. Similar vascular abnormalities have been found in other muscular dystrophies, including Duchenne muscular dystrophy, where reduced capillary density contributes to tissue hypoxia and exacerbates muscle damage [42]. Therefore, the ability of Boldo to restore vascularization could represent an additional mechanism by which it protects skeletal muscles in dysferlinopathy.

Lower physical performance due to this disease is associated with muscle atrophy, characterized by a decrease in the CSA of muscle fibers [33,43]. Notably, the use of Boldo reverses the CSA of skeletal myofibers in blAJ mice. Additionally, an important histopathological feature of dysferlinopathy is the presence of centrally located nuclei within muscle fibers, indicative of ongoing cycles of degeneration and regeneration [44]. We found that Boldo treatment reduces the number of centrally nucleated myofibers in blAJ mice, suggesting a reduction in continuous cell death and regeneration processes.

Extensive fatty infiltration of myofibers in patients with dysferlinopathy has also been observed through imaging techniques such as magnetic resonance imaging [45]. In one reported case of limb–girdle muscular dystrophy type 2B due to dysferlin deficiency, the lumbar and lower thoracic portions of the erector spinae muscles were observed to be entirely filled with fatty vesicles [46]. The same occurs in animal models such as A/Jdys^−/−^ and blAJ, for which lipid accumulation in skeletal muscles was reported [34]. Interestingly, in this study, we found that treatment with Boldo decreased lipid accumulation in skeletal muscles from blAJ mice. Given that this fat accumulation does not occur in blAJ mice with myofibers deficient in Cx43 and Cx45 [33], a proposed explanation for lipid accumulation is the propensity of muscle myogenic cells to adopt an adipogenic phenotype. Accordingly, a human dysferlinopathy myoblast cell line, cultured under conditions that promote myogenic differentiation, has been shown to manifest aberrant adipogenic commitment, as indicated by PPARγ expression [30]. This phenomenon accounts for the increased number of oil red O-positive cells and the reduced formation of myotubes observed at later stages of differentiation. Here, again, treatment with boldine effectively reversed aberrant adipogenic commitment, reducing PPARγ expression, decreasing lipid accumulation, and restoring myotube formation capacity [30].

In addition to the effect of lower lipid accumulation on progenitor cell fate, it may also result in the progressive degeneration and removal of previously lipid-laden muscle fibers [47]. As these damaged fibers are replaced by newly regenerated myofibers formed under conditions of improved perfusion and lower inflammation, the new muscle tissue progressively eliminates the aberrant lipid content, thereby contributing to the overall decrease in intramuscular fat observed after Boldo treatment.

Greater sarcolemma permeability observed in muscle fibers from blAJ mice would be mediated, at least in part, by the expression of large-pore channels such as Cx43 and Cx45 hemichannels. Previous studies have shown that blAJ mice deficient in Cx43 and Cx45 do not exhibit elevated sarcolemma permeability, indicating that the absence of these connexins is sufficient to mitigate the rise in membrane permeability [33]. Here, treatment with Boldo was found to cause a similar effect on sarcolemma permeability of myofibers from bIAJ mice. These findings parallel the results observed in myofibers from bIAJ mice deficient in Cx43 and Cx45 expression, suggesting that Boldo may exert its protective action by modulating the activity and/or expression of these non-selective channels.

Cx43 HCs, P2X7 receptors, and the TRPV2 channel have all been shown to be present in muscular biopsies from dysferlinopathy patients [31] and are permeable to Ca^2+^ [48,49,50]. Thus, they can all contribute to sustained increases in cytoplasmic Ca^2+^ levels [31,33]. Additionally, Cx HCs mediate the efflux of ATP into the extracellular space [51], where it can activate ionotropic purinergic receptors (P2X receptors), resulting in further Ca^2+^, and constituting a feedforward mechanism that activates the inflammasome complex [52]. This sequence of events promotes a chronic inflammatory state, which is a well-established feature of dysferlinopathy [53,54,55].

Supporting the relevance of inflammasome activation in dysferlinopathy, previous studies have shown increased levels of NLRP3, ASC, active caspase-1, and mature IL-1β in skeletal muscles from dysferlin-deficient (A/J) mice [36]. Myotubes from primary myoblast cultures of these animals have also been shown to express these components, confirming that muscle cells alone can assemble a functional inflammasome and secrete IL-1β, a key cytokine driving chronic inflammation [36]. In line with these reports, we also observed elevated expressions of inflammasome-related components, including ASC, NLRP3, and IL-1β, as determined by qPCR analysis in skeletal muscle from blAJ mice. Interestingly, Boldo treatment significantly reduced the levels of these markers, further supporting its role in dampening inflammasome-mediated inflammation in dysferlinopathy.

In conclusion, our findings indicate that Boldo improves vascular and muscle integrity and supports its potential as a complementary therapeutic strategy for dysferlinopathy. Building on these results, we highlight Boldo’s potential as a therapeutic agent for dysferlinopathy, contingent upon standardized dosage, administration, and treatment duration. The animals included in this study were symptomatic blAJ mice, which reinforces the translational relevance of our findings. We advocate for future clinical studies to evaluate Boldo’s therapeutic potential, which is also supported by its lack of toxicity and its use in clinical trials for treating overactive bladder in women (National Library of Medicine).

## 4. Materials and Methods

### 4.1. Reagents

Ethidium (Etd^+^) bromide, DMEM/F12 medium, and FBS were purchased from GIBCO/BRL (Grand Island, NY, USA). Fluoromount g and 4′,6-diamidino-2-fenilindol (DAPI) were obtained from Science (Hatfield, PA, USA).

### 4.2. Animals

This design included wild-type C57Bl/6 and blAJ male mice (dysferlin-deficient animals). These animals bear a homozygous mutation in both alleles of the DYSF gene and present several muscle alterations representative of this type of muscular dystrophy. All protocols were approved by the Bioethics Committee of the Universidad de Valparaíso, Chile (CBC 143-2025), in accordance with the ethical standards established in the 1964 Declaration of Helsinki and its later amendments. All efforts were made to minimize animal suffering and reduce the number of animals used, and alternatives to in vivo techniques were implemented when possible. This study was designed as a pilot study; therefore, no formal power calculation was performed. A sample size of *n* = 4 per group was chosen based on previous experience with the blAJ model and recommendations for exploratory studies [56].

### 4.3. Boldo Treatment

The different mouse strains were treated with pulverized Boldo (*Peumus boldus*) leaves (50 mg/kg, daily) for 4 weeks. The leaves were from the Chilean tea brand (Té Supremo) and were administered to mice mixed with 300 mg of peanut butter in a separate cage (one mouse at a time).

### 4.4. Blood Perfusion

In vivo muscle perfusion was assessed by using the Pericam^®^ PSI-HR system (Perimed Ltd., Stockholm, Sweden). Adult mice were anesthetized with 4% isoflurane, and the skin over the right gastrocnemius muscle was carefully removed to expose the microvasculature. Microvascular blood flow was recorded from four predefined circular regions of interest (ROIs) within the muscle, consistently positioned across animals to capture representative microcirculation.

The experimental protocol included a 3–4 min baseline perfusion recording, followed by a single bolus of acetylcholine (10 µM) to induce endothelium-dependent vasodilation. Continuous perfusion monitoring was maintained throughout the experiment (~10 min). For quantification, time-of-interest (TOI) windows of 40 s were extracted for both basal and post-acetylcholine conditions, beginning immediately after drug administration. Perfusion values are reported in raw perfusion units.

Data acquisition and analysis were conducted by two independent investigators (F.T. and H.S.). To minimize bias, experiments were performed in a randomized, head-to-head fashion, simultaneously including mice from all experimental groups. Data analysis was performed in a blinded manner.

### 4.5. Muscular Force Test

The four-limb suspension test was used to assess muscle strength in mice. Animals were placed on a wire grid at a height of 35 cm, with a soft bed underneath to prevent injury in case of a fall. Suspension time was recorded in seconds. Each mouse performed several trials, with 2 min rest intervals between each trial.

### 4.6. Histology

Muscle samples were fixed with 4% paraformaldehyde for 48 h at 4 °C. Subsequently, those samples were dehydrated in alcohol baths and then embedded in paraffin. Sections of 5 μm thickness were cut using a microtome (Accu-Cut^®^ SRM™ 200, Sakura Finetek, Torrance, CA, USA). Finally, histological staining with Hematoxylin and Eosin (H&E) was performed to observe variations in tissue architecture. Masson’s Trichrome staining was used for collagen fiber analysis, for which nuclear staining was performed with Weigert’s hematoxylin, followed by a Biebrich Scarlet solution (acidic fuccina at 1%, Biebrich Scarlet at 1%, and glacial acetic acid) for cytoplasmic staining for 10 s. Then, a mordant solution of phosphotungstic/phosphomolybdic acid was applied for 15 min. Finally, to highlight the collagen fibers, an aniline blue solution (aniline blue, acetic acid, and distilled water) was used. All images were captured with a 40×objective under an optical microscope (Olympus CX43RF, Olympus Corporation, Tokyo, Japan) equipped with a digital camera (Euromex 4K Sony Ultra HD VC.3040, Euromex Microscopes, Arnhem, The Netherlands).

### 4.7. Cross-Sectional Area

This parameter was evaluated in the tibialis anterior (TA) muscle fibers. Muscles were fixed in 4% (wt/vol) paraformaldehyde, fast frozen in isopentane, and maintained in liquid nitrogen. Then, the muscles were oriented perpendicular and mounted in OCT. Cross-sections (10 µm thick) of frozen muscles were obtained using a cryostat (Leica CM1100, Leica Microsystems, Wetzlar, Germany) and ImageJ software (version 1.54f; National Institutes of Health, Bethesda, MD, USA).

### 4.8. Immunofluorescence Analysis

Muscles were fast frozen with liquid nitrogen. Then, cross-sections (12 μm thick) were obtained by using a cryostat and fixed with 4% formaldehyde for 10 min at room temperature. Sections were incubated for 3 h at room temperature in blocking solution (50 mM NH_4_Cl, 0.025% Triton, 1% BSA on PBS solution 1×), incubated overnight with appropriate dilutions of primary antibody directed to CD31, washed five times with PBS 1× solution, followed by 1 h incubation with secondary rabbit antibody conjugated to Cy2 or Cy3, purchased from Jackson ImmunoResearch Laboratories (West Grove, PA, USA). They were then mounted in Fluoromount G obtained from Electron Microscopy Science (Hatfield, PA, USA). Immunoreactive binding sites were localized under a Nikon Eclipse Ti microscope (Nikon Corporation, Tokyo, Japan) equipped with epifluorescence illumination.

### 4.9. Oil Red O Staining

This staining was carried out as previously described [57]. Briefly, cross-sections of gastrocnemius muscles mounted on glass slides were fixed with 4% paraformaldehyde in the presence of 180 mM CaCl_2_. Subsequently, oil red O reagent was added, and the excess was removed after 30 min, after which the samples were rinsed with water. We performed quantitative analysis of oil red O staining by calculating the stained area fraction using Python 3.12.11 OpenCV (version 4.9.0), NumPy (version 1.26.4), and Matplotlib libraries (version 3.9.2).

### 4.10. Isolation of Mouse Skeletal Myofibers

Myofibers were isolated from flexor digitorum brevis (FDB) muscles as previously described [40]. FDB muscles were carefully dissected and immersed in culture medium (DMEM/F12 supplemented with 10% FBS) containing 0.2% collagenase type I, incubated for 3 h at 37 °C, and transferred to a 15 mL test tube (Falcon) with 5 mL of culture medium. Then, muscle tissue was gently triturated 10 times by using a Pasteur pipette with a wide tip to disperse single myofibers. Dissociated myofibers were centrifuged at 1000 rpm for 15 s (model 8700 centrifuge; Kubota, Tokyo, Japan) and washed twice by sedimentation, first in PBS solution and then in Krebs buffer (in mM: 145 NaCl, 5 KCl, 3 CaCl_2_, 1 MgCl_2_, 5.6 glucose, 10 HEPES-Na, pH 7.4), the latter containing 10 μM BTS to inhibit contractions and reduce myofiber damage during the isolation procedure. Finally, fibers were suspended in 5 mL of Krebs HEPES buffer with 10 μM BTS, plated in plastic culture dishes or placed in 1.5 mL microcentrifuge tubes, and kept at room temperature.

### 4.11. Time-Lapse Recording of Etd^+^ Uptake

Cellular uptake of Etd^+^ was evaluated by time-lapse measurements as previously described [40]. Briefly, freshly isolated myofibers plated onto plastic culture dishes were washed twice with Krebs buffer solution containing 5 μM Etd^+^. Etd^+^ fluorescence intensity was recorded in regions of interest that corresponded to myofiber nuclei by using a conventional Nikon Eclipse Ti fluorescent microscope.

### 4.12. Reverse Transcription Polymerase Chain Reaction (PCR)

Total RNA was isolated from skeletal muscle using TRIzol following the manufacturer’s instructions (Invitrogen, Waltham, MA, USA). Two microgram aliquots of total RNA were transcribed to cDNA using MMLV-reverse transcriptase (Fermentas, Waltham, MA, USA), and mRNA levels were evaluated by PCR amplification (GoTaq Flexi DNA polymerase; Promega, Madison, WI, USA).

### 4.13. The Oligos Used Were the Following

NLRP3: S 5′-GCTGGCATCTGGGGAAACCT-′3, AS 5′GCCCTTCTGGGGAGGATAGT-′3; IL-1β: S 5′-TGG GAT GAT GAT GAT AAC CT-3′, AS 5′-CCC ATA CTT TAG GAA GAC AGG GAT TT-3′; 18S S 5′-TCAAGAACGAAAGTCGGAGG-′3, AS 5′-GGACATCTAAGGGCATCACA-′3. All reactions were performed with an initial denaturation of 5 min at 95 °C, followed by 40 cycles of 30 s at 95 °C, and annealing for 30 s at 60 °C in a PikoReal™ Real-Time PCR System (ThermoFisher Scientific™, Waltham, MA, USA). The C_t_ values of each cDNA resulting from three different experiments were normalized using the 2^−ΔΔCt^ method, with 18S serving as the reference gene.

### 4.14. Statistical Analysis

Quantitative variables are presented as median ± standard deviation. Considering data distribution, we used parametric or non-parametric tests, as appropriate, using the Shapiro–Wilk normality test. For multiple group comparisons during perfusion analysis, data are presented as mean values calculated from four different subjects. Multiple comparisons were performed using ordinary one-way ANOVA, followed by multiple comparisons by controlling the false discovery rate by using the Benjamini and Hochberg method. In other experiments, data were analyzed using one-way ANOVA followed by Tukey’s multiple-comparison test and an appropriate normality test. *p* < 0.05 was considered a statistically significant difference. Data and statistical analyses were performed using the Microsoft Excel database and GraphPad Prism 6 (GraphPad Software, La Jolla, CA, USA).

## Figures and Tables

**Figure 1 ijms-26-09945-f001:**
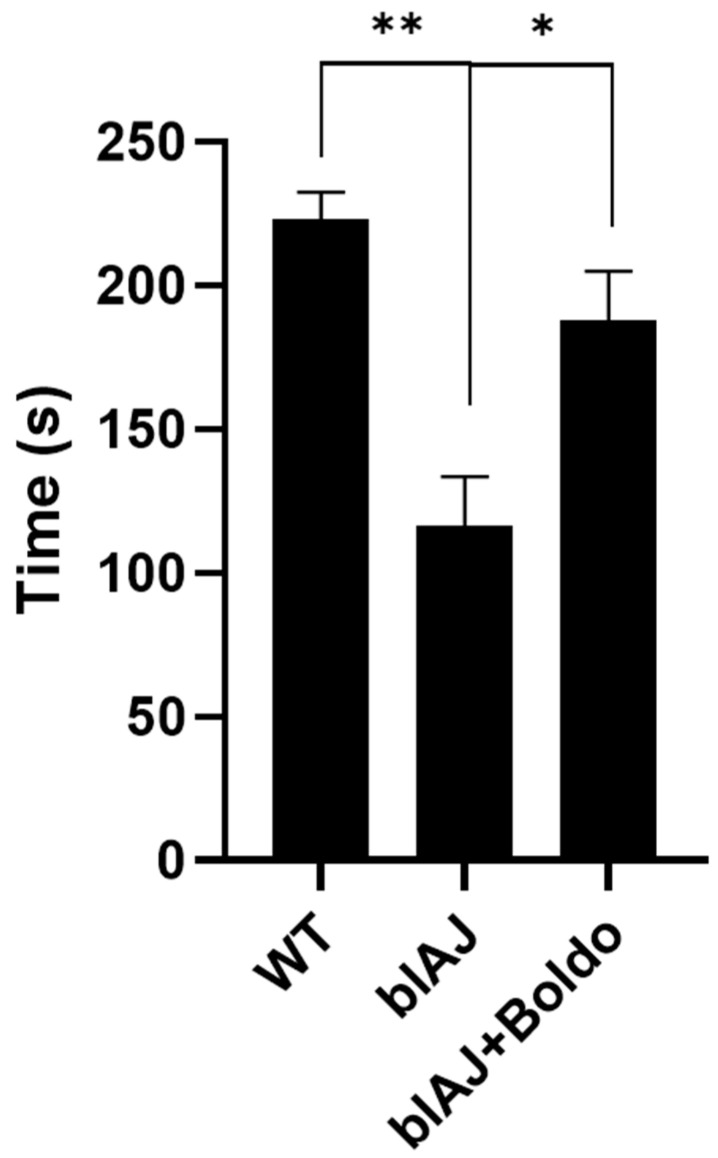
Boldo recovers physical performance in symptomatic blAJ mice. Physical performance was evaluated using the four-limb suspension test: physical performance was assessed in three groups of animals: wild-type (WT), bIJA, and Boldo-treated bIJA mice (bIJA + Boldo). *n* = 4. ** *p* < 0.01, * *p* < 0.05, Tukey test.

**Figure 2 ijms-26-09945-f002:**
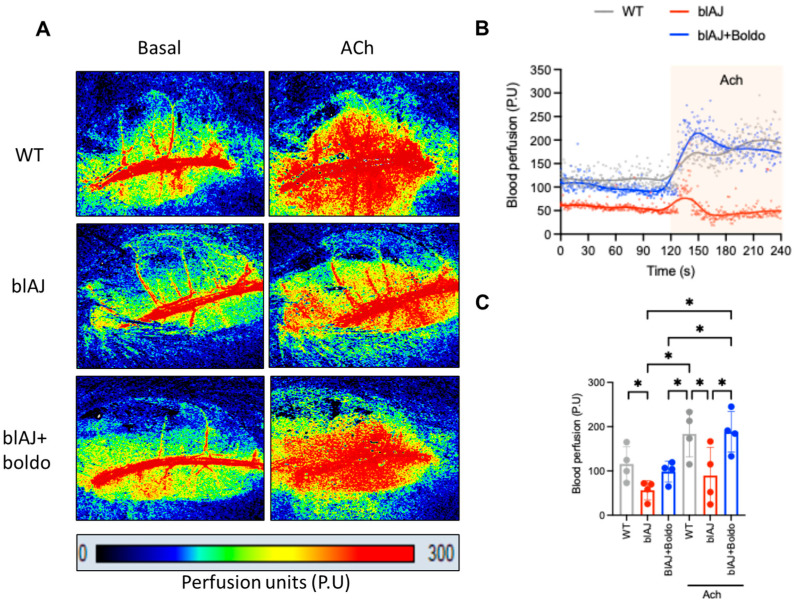
Blood perfusion: (**A**) Representative images of perfusion analysis using the Pericam^®^ PSI-HR system in the absence (Basal) or in response to acetylcholine (Ach, 10 μM, topical application) in adult wild-type (WT) (WT, gray), blAJ (red), and Boldo-treated blAJ (blue) mice. Perfusion units (arbitrary color units) from 0 to 300. (**B**) Representative trace of blood perfusion from the three experimental groups. The marked area indicates experiments in the presence of Ach. (**C**) Average of four experiments in individual subjects treated as in (**A**). Each dot represents a randomly selected individual subject. Data are presented as median ± standard deviation. * *p* < 0.05. One-way ANOVA. Multiple comparisons by controlling the false discovery rate.

**Figure 3 ijms-26-09945-f003:**
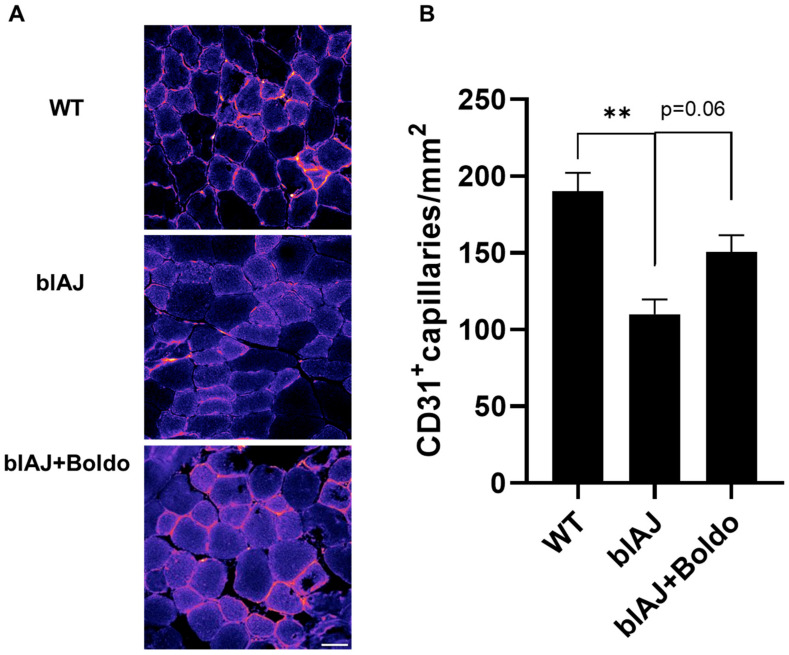
Boldo recovers the capillary density of skeletal muscles from blAJ mice: (**A**) Representative immunofluorescence images of tibialis anterior muscle cross-sections from wild-type (WT), blAJ, and Boldo-treated blAJ mice stained for CD31 (red), an endothelial cell marker. Scale bar: 50 μm. (**B**) Quantification of CD31^+^ capillaries per mm^2^ reveals a reduced capillary density in blAJ mice compared to WT, which is partially restored by Boldo treatment. Data are presented as mean ± SEM. *n* = 4. ** *p* < 0.01. Tukey test.

**Figure 4 ijms-26-09945-f004:**
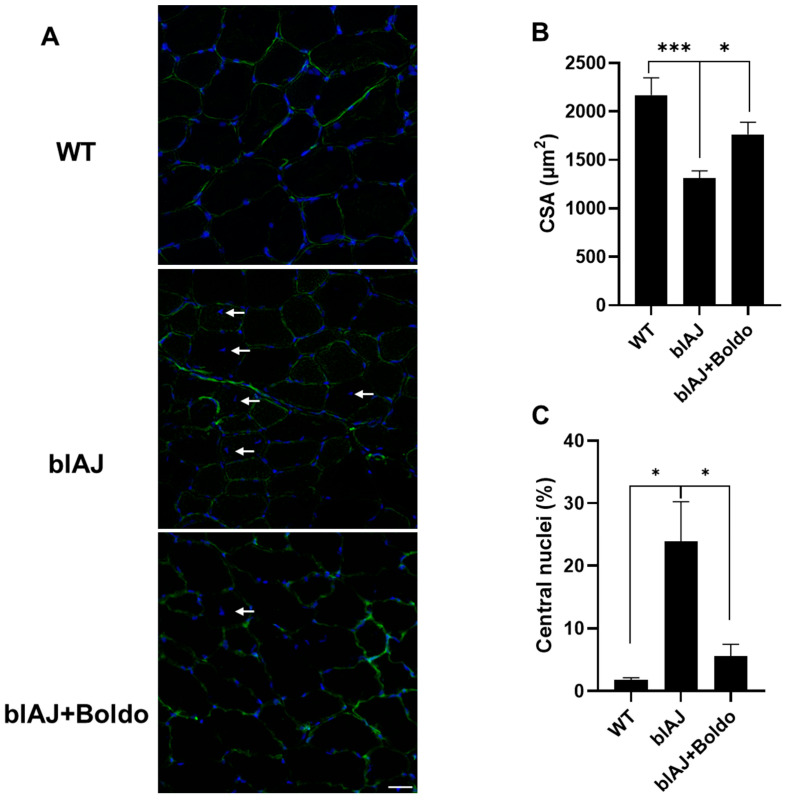
Boldo recovers the cross-sectional area of *tibialis anterior* myofibers of blAJ mice: (**A**) Distribution of DAPI-stained nuclei (blue) is shown in the middle panel, indicated by white arrowheads. Fiber membranes are labeled with WGA (green). Scale bar: 50 μm. (**B**) Quantification of cross-sectional area (CSA). *n* = 4. *** *p* < 0.001, * *p* < 0.05. Tukey test. (**C**) Quantification of the internal nuclei index obtained from DAPI-stained muscle cross-sections. * *p* < 0.05. Tukey test.

**Figure 5 ijms-26-09945-f005:**
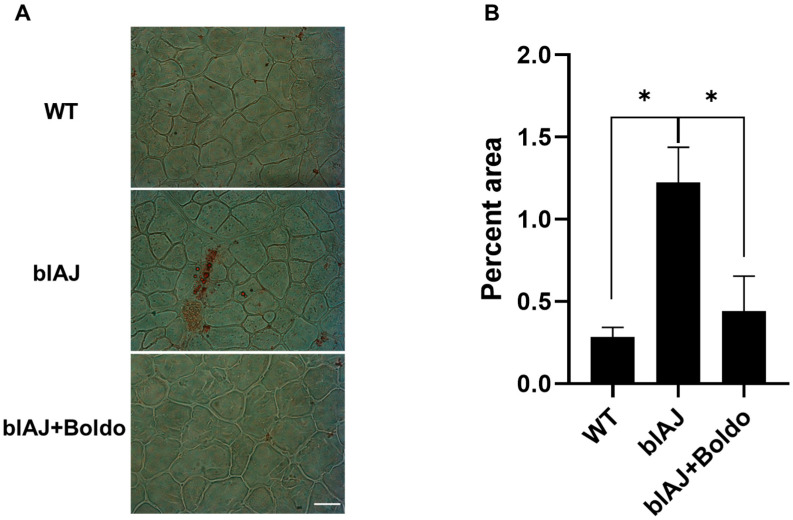
Boldo reduces lipid accumulation in skeletal muscles from blAJ mice: (**A**) Cryosections of gastrocnemius muscles from wild-type (WT), blAJ, and Boldo-treated blAJ mice were analyzed for lipid accumulation using oil red O staining. Scale bar: 50 μm. (**B**) Quantification of the percentage of oil red O-positive area per field. Data represent mean ± SEM (*n* = 4 mice per group). * *p* < 0.05, Tukey’s test.

**Figure 6 ijms-26-09945-f006:**
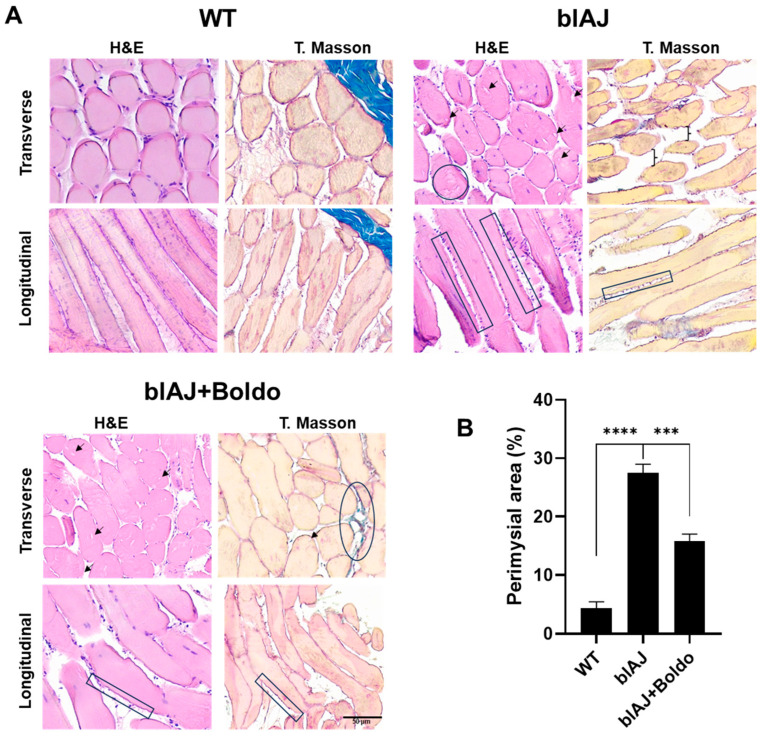
Histopathological analysis of skeletal muscles from WT, blAJ, and blAJ mice treated with Boldo: (**A**) Representative transverse (top row) and longitudinal (bottom row) sections of tibialis anterior muscle stained with hematoxylin and eosin (H&E, left column) or Masson’s trichrome (right column). WT muscle displays uniform fiber diameter, peripheral nuclei, and preserved endomysia and perimysium connective tissue. Muscles from blAJ mice showed dystrophic changes, including fiber size variability, central nuclei, vacuolization (**→**), necrotic fibers (◯), sarcolemma projections resembling microvilli (⫿), and epimysia separation (}), along with increased collagen deposition (Masson). Scale bar: 50 μm. (**B**) Quantification of perimysial area expressed as a percentage of section area. Data are mean ± SEM (*n* = 4). **** *p* < 0.0001, *** *p* < 0.001. Tukey’s test.

**Figure 7 ijms-26-09945-f007:**
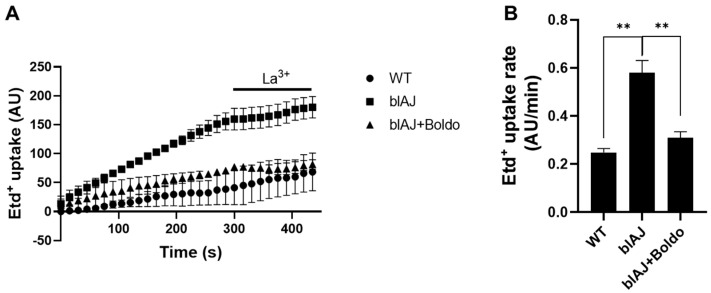
Elevated sarcolemma permeability of freshly isolated blAJ muscle fibers is recovered by Boldo treatment in blAJ mice: (**A**) Representative curve of Etd^+^ uptake over time in isolated myofibers from FDB muscles, *n* = 4 independent experiments; six myofibers were recorded in each experiment, and each value is the mean ± SEM. (**B**) Etd^+^ uptake rate obtained from slopes of curves similar to A. Values represent means ± SEM. *n* = 4. ** *p* < 0.01, Tukey test.

**Figure 8 ijms-26-09945-f008:**
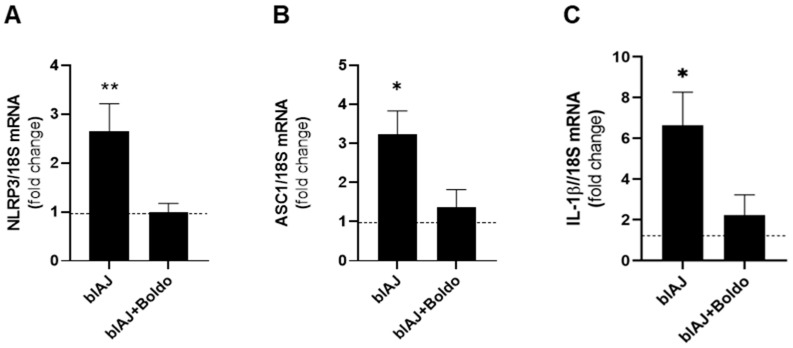
Boldo treatment reduces mRNA levels of inflammasome components in blAJ mice. Total RNA was isolated from skeletal muscles and mRNA levels of (**A**) NLRP3, (**B**) ASC, and (**C**) IL-1β were evaluated by qPCR. *n* = 4. ** *p* < 0.01, * *p* < 0.05 vs. control, Tukey test Dashed line indicates the mean control value.

## Data Availability

The data supporting the conclusion of this article will be made available by the authors, without undue reservation. The data can be found in https://figshare.com/articles/figure/_/30234910 (accessed on 11 October 2025).

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
