# Peer review of "Boldo Restores Vascularization and Reduces Skeletal Muscle Inflammation in Symptomatic Mice with Dysferlinopathy"

_ijms, 2025, doi:10.3390/ijms26209945_

Round 1
Reviewer 1 Report
Comments and Suggestions for Authors
Reviewer’s Comments – General Overview
This manuscript presents a preclinical evaluation of pulverized Peumus boldus (Boldo) leaves as a potential nutraceutical intervention for dysferlinopathy, using the blAJ mouse model. The study addresses a significant unmet medical need, as no approved therapies currently exist to halt the progression of dysferlinopathies. The authors report improvements in muscle function, vascular performance, histological integrity, and inflammatory markers following Boldo treatment, suggesting a multifaceted therapeutic effect.
Overall, the topic is timely and relevant, and the experimental approach is of interest. However, several aspects of the manuscript require clarification, refinement, and further justification to strengthen the scientific rigor and interpretability of the findings. Below, I provide detailed comments and suggestions.
- Experimental Design: Choice of Boldo Leaves vs. Boldine
The manuscript highlights the presence of several bioactive compounds in Peumus boldus leaves, including alkaloids such as boldine, flavonoids, phenolic acids, and essential oils, which are known to confer antioxidant and anti-inflammatory properties. Given that boldine has been specifically reported to exert protective effects in inflammatory models—potentially through the inhibition of connexin and pannexin hemichannels—it would be helpful for the authors to clarify their rationale for using pulverized Boldo leaves rather than purified boldine in this study.
Was the decision based on the potential synergistic effects of the full phytocomplex, the accessibility or regulatory status of the whole leaf preparation, or other considerations such as traditional use or bioavailability? A brief justification would strengthen the interpretation of the results and help contextualize the translational relevance of the chosen formulation.
- Muscle Regeneration Mechanisms
The manuscript reports that Boldo treatment reduces muscle fiber degeneration and improves tissue architecture in blAJ mice. Since satellite cells (MuSCs) are essential for muscle regeneration (Dumont et al., 2015 (https://doi.org/10.1002/j.2040-4603.2015.tb00646.x), it would be valuable to know whether the authors considered their cell dynamics in their analysis, since MuSCs disfunction contribute to this and other related pathologies, such as Duchenne (Kodippili and Rudnicki, 2023; https://doi.org/10.3389/fphys.2023.1180980). This recent study from Escobar et al., 2025 (https://doi.org/10.1038/s41467-024-55086-0) demonstrates that gene-edited primary MuSCs from patients and humanized mouse models can restore dysferlin expression and regenerate muscle tissue, confirming that impaired MuSC function contributes significantly to disease pathology.
Did the authors assess markers of MuSCs activation or proliferation (e.g., Pax7, MyoD, Ki67) to determine whether Boldo influences regenerative capacity through modulation of these cells? If not, do they plan to explore this in future studies? Including such data or discussing this possibility could provide deeper insight into the mechanisms underlying the observed histological improvements.
- Histological Analysis
The manuscript provides a descriptive evaluation of muscle fiber architecture using H&E and Masson’s Trichrome staining, highlighting key structural differences between groups. While these observations are informative, the analysis remains qualitative. It would strengthen the study considerably if the authors included quantitative data on histological alterations—such as fiber cross-sectional area distribution, percentage of centrally nucleated fibers, collagen deposition, or vacuole frequency. Could the authors clarify why a quantitative morphometric analysis was not performed? If such data are available, including them would enhance the robustness of the conclusions. If not, a brief discussion of this limitation and plans to address it in future studies would be appreciated.
Other comments:
- Line 150: Replace the semicolon with a comma before “we evaluated.”
- Line 166: Replace the period with a comma before “We decided.”
- Line 170: Refer to “Figure 5” instead of “Figure 5A,” as the figure contains only one panel with three images.
- Figure 6: Please cite this figure explicitly in the text to support the histological observations.
- Line 227: “Figure 7B” should be corrected to “Figure 8B.”
- Line 244: Add a comma after the citation [37,38].
- Boldo treatment: The manuscript states that Boldo was administered as “3 mg peanut butter”. This dosage appears unusually low for a nutraceutical intervention. Please, revise.
- Institutional Review Board Statement, Data Availability Statement, and Abbreviations: These sections should be reviewed and completed.
Author Response
|
Comments 1: Experimental Design: Choice of Boldo Leaves vs. Boldine. The manuscript highlights the presence of several bioactive compounds in Peumus boldus leaves, including alkaloids such as boldine, flavonoids, phenolic acids, and essential oils, which are known to confer antioxidant and anti-inflammatory properties. Given that boldine has been specifically reported to exert protective effects in inflammatory models—potentially through the inhibition of connexin and pannexin hemichannels—it would be helpful for the authors to clarify their rationale for using pulverized Boldo leaves rather than purified boldine in this study. Was the decision based on the potential synergistic effects of the full phytocomplex, the accessibility or regulatory status of the whole leaf preparation, or other considerations such as traditional use or bioavailability? A brief justification would strengthen the interpretation of the results and help contextualize the translational relevance of the chosen formulation.
|
|
Response 1: We appreciate the reviewer’s comment. Our rationale for using powdered Boldo leaves rather than purified boldine is based on translational accessibility. Boldo is traditionally consumed as an herbal infusion in South America as well as in some European countries like Spain and Italy and is already available worldwide (e.g., available in Amazon) as a dietary supplement or complementary therapy, which does not require FDA approval. In contrast, boldine, although recognized as one of the major alkaloids of Boldo, has not been approved by the FDA as a therapeutic agent and, to our knowledge, has not yet been tested in human clinical studies. Therefore, the use of Boldo leaves provides a more feasible and accessible strategy for potential translation, while still delivering boldine together with other bioactive compounds that may contribute to the beneficial effects observed. This indicated in the introduction and Discussion sections.
Comments 2: Muscle Regeneration Mechanisms The manuscript reports that Boldo treatment reduces muscle fiber degeneration and improves tissue architecture in blAJ mice. Since satellite cells (MuSCs) are essential for muscle regeneration (Dumont et al., 2015 (https://doi.org/10.1002/j.2040-4603.2015.tb00646.x), it would be valuable to know whether the authors considered their cell dynamics in their analysis, since MuSCs disfunction contribute to this and other related pathologies, such as Duchenne (Kodippili and Rudnicki, 2023; https://doi.org/10.3389/fphys.2023.1180980). This recent study from Escobar et al., 2025 (https://doi.org/10.1038/s41467-024-55086-0) demonstrates that gene-edited primary MuSCs from patients and humanized mouse models can restore dysferlin expression and regenerate muscle tissue, confirming that impaired MuSC function contributes significantly to disease pathology. Did the authors assess markers of MuSCs activation or proliferation (e.g., Pax7, MyoD, Ki67) to determine whether Boldo influences regenerative capacity through modulation of these cells? If not, do they plan to explore this in future studies? Including such data or discussing this possibility could provide deeper insight into the mechanisms underlying the observed histological improvements.
Response 2: We agree with the Reviewer that MuSC dysfunction contributes significantly to dysferlinopathy. In the present study, we did not evaluate markers of MuSC activation or proliferation (such as Pax7, MyoD, or Ki67), since our focus was on functional, vascular, and inflammatory outcomes. We acknowledge this as a limitation and have added it to the Discussion. Nevertheless, Cea et al. (2020) provided experimental evidence that boldine favors the differentiation of dysferlin-mutant myoblasts into multinucleated myotubes, as evidenced by MHC immunoreactivity and increased nuclei per myotube. Although in our study we used Boldo leaves and not purified boldine, this finding supports the possibility that inhibition of connexin hemichannels may also promote regenerative processes at the cellular level.
Comments 3: Histological Analysis The manuscript provides a descriptive evaluation of muscle fiber architecture using H&E and Masson’s Trichrome staining, highlighting key structural differences between groups. While these observations are informative, the analysis remains qualitative. It would strengthen the study considerably if the authors included quantitative data on histological alterations—such as fiber cross-sectional area distribution, percentage of centrally nucleated fibers, collagen deposition, or vacuole frequency. Could the authors clarify why a quantitative morphometric analysis was not performed? If such data are available, including them would enhance the robustness of the conclusions. If not, a brief discussion of this limitation and plans to address it in future studies would be appreciated.
Response 3: We thank the Reviewer for this valuable comment. Quantitative morphometric analyses were already included in the manuscript: myofiber cross-sectional area (CSA) is presented in Figure 4B and the percentage of centrally nucleated fibers in Figure 4C. As expected for dysferlinopathy, CSA was significantly reduced in blAJ mice compared to controls, and Boldo treatment during only short period of time partially but significantly restored CSA. Likewise, blAJ mice showed a significant increase in centrally nucleated fibers, whereas Boldo treatment reduced this parameter to levels significantly lower than untreated blAJ mice.
|
Other comments:
- Line 150: Replace the semicolon with a comma before “we evaluated.”
- Line 166: Replace the period with a comma before “We decided.”
- Line 170: Refer to “Figure 5” instead of “Figure 5A,” as the figure contains only one panel with three images.
- Figure 6: Please cite this figure explicitly in the text to support the histological observations.
- Line 227: “Figure 7B” should be corrected to “Figure 8B.”
- Line 244: Add a comma after the citation [37,38].
- Boldo treatment: The manuscript states that Boldo was administered as “3 mg peanut butter”. This dosage appears unusually low for a nutraceutical intervention. Please, revise.
- Institutional Review Board Statement, Data Availability Statement, and Abbreviations: These sections should be reviewed and completed.
Response: We thank the Reviewer for these helpful observations. All suggested editorial corrections have been implemented in the revised manuscript. In addition, we corrected a typographical error regarding the Boldo dose: the correct dose is 300 mg, not 3 mg as originally written.
Reviewer 2 Report
Comments and Suggestions for Authors
Thanks to the authors for conducting this study
The authors are studying the potential protective effect of Boldo which is a naturally extracted herbal in ameliorating the skeletal muscle phenotype found in dysferlinopathy.
The study aim and question were clear, the author aimed to explore the effect of Boldo treatment on muscle function and vascularity, also they assessed the inflammatory infiltration in the Boldo treated mice compared to controls.
In figure 3 the authors counted the CD31+ cells and normalized to the myofibers, The endothelial cells are located in the endomysium in between the myofibers not inside. The quantification should be done on the whole TA section and presented in number per CSA. Also, the images have a super bright blue background, are you increasing the exposure of the DAPI channel? please clarify.
Please include the scale bar size for images in figure 3 and 4.
Figure 4, the DAPI staining is nice. Regarding the fiber CSA, are you quantifying the CSA in selected images or in the whole muscle section? Please can you present the data in frequency distribution of myofibers in different rages of sizes.
Figure 5 the oil red staining is not clear, and I see that the WT mice have more than the bIAJ+Boldo, please quantify and retake the images with better brightness.
In figure 6 please revise the images highlights, I did only noticed the arrow heads and the other shapes but not the dots which should aim at necrotic fibers. Please do quantification.
In figure 8 the Y axis of the PCR result is labeled “fold increase” however the treatment decrease the expression of inflammatory markers, Please change in to “fold change”
Author Response
Comments and Suggestions for Authors
Thanks to the authors for conducting this study
The authors are studying the potential protective effect of Boldo which is a naturally extracted herbal in ameliorating the skeletal muscle phenotype found in dysferlinopathy.
The study aim and question were clear, the author aimed to explore the effect of Boldo treatment on muscle function and vascularity, also they assessed the inflammatory infiltration in the Boldo treated mice compared to controls.
In figure 3 the authors counted the CD31+ cells and normalized to the myofibers, The endothelial cells are located in the endomysium in between the myofibers not inside. The quantification should be done on the whole TA section and presented in number per CSA. Also, the images have a super bright blue background, are you increasing the exposure of the DAPI channel? please clarify.
Response: We thank the reviewer for this important observation. We agree that endothelial cells (CD31+) are located in the endomysium and not inside myofibers. Following the reviewer’s suggestion, we have re-analyzed our data and now present CD31+ capillary density expressed as the number of CD31+ structures per square millimeter of muscle cross-sectional area (CSA). As we did not capture entire muscle sections, we quantified several non-overlapping random fields per section and calculated the average CD31+ density.
Please include the scale bar size for images in figure 3 and 4.
Response: We thank the Reviewer for this comment. Scale bars have now been added to Figures 3 and 4, and their sizes are indicated in the corresponding legends.
Figure 4, the DAPI staining is nice. Regarding the fiber CSA, are you quantifying the CSA in selected images or in the whole muscle section? Please can you present the data in frequency distribution of myofibers in different rages of sizes.
Response: We thank the Reviewer for this comment. CSA was quantified from several representative fields covering different regions of each TA muscle, and this has now been clarified in Methods. We did not perform frequency distribution analysis of CSA. Our results showed that CSA was significantly reduced in blAJ mice compared to WT controls, consistent with previous reports in dysferlinopathy, and that Boldo treatment partially but significantly improved this parameter after a short period of treatment.
Figure 5 the oil red staining is not clear, and I see that the WT mice have more than the bIAJ+Boldo, please quantify and retake the images with better brightness.
Response: We performed quantitative analysis of Oil Red O staining by calculating the stained area fraction using Python 3.12.11 OpenCV, NumPy, and Matplotlib libraries. These results are now included as a bar graph in Figure 5B, complementing the representative images.
In figure 6 please revise the images highlights, I did only noticed the arrow heads and the other shapes but not the dots which should aim at necrotic fibers. Please do quantification.
Response: We thank the Reviewer for this comment. CSA was quantified from several representative fields covering different regions of each TA muscle, and this has now been clarified in Methods. We did not perform frequency distribution analysis of CSA. Our results showed that CSA was significantly reduced in blAJ mice compared to WT controls, consistent with previous reports in dysferlinopathy, and that Boldo treatment partially but significantly improved this parameter after a short period of treatment.
In figure 8 the Y axis of the PCR result is labeled “fold increase” however the treatment decrease the expression of inflammatory markers, Please change in to “fold change”
Response: We thank the Reviewer for noticing this mistake. We have corrected the Y axis label to “fold change” in Figure 8.
Round 2
Reviewer 1 Report
Comments and Suggestions for Authors
The authors have responded to all the questions raised. The text now meets the required quality for publication. Nevertheless, they should review how the figures are cited in the text, as some of the figure panels are not referenced. In such cases, it might be better to cite only the figure without specifying the panel.
Author Response
We thank the reviewer for the positive evaluation of our revised manuscript. Following the suggestion, we carefully reviewed all figure citations in the text. We corrected inconsistencies and, where some panels were not referenced, we now cite the entire figure as recommended.
Reviewer 2 Report
Comments and Suggestions for Authors
Thanks.
Author Response
We thank the reviewer for the supportive comment.